# A Randomized Prospective Non-Inferiority Trial of Sentinel Lymph Node Biopsy in Early Breast Cancer: Blue Dye Compared with Indocyanine Green Fluorescence Tracer

**DOI:** 10.3390/cancers14040888

**Published:** 2022-02-10

**Authors:** Michel Coibion, Fabrice Olivier, Audrey Courtois, Nathalie Maes, Véronique Jossa, Guy Jerusalem

**Affiliations:** 1Gynecology Department, CHC Montlegia, 4020 Liege, Belgium; josscoi@hotmail.com; 2Medical Oncology Department, University Hospital of Liege, 4020 Liege, Belgium; audrey.courtois@chc.be (A.C.); g.jerusalem@chuliege.be (G.J.); 3Biostatistics and Medico-Economic Information Department, University Hospital of Liège, 4020 Liege, Belgium; nmaes@chuliege.be; 4Anatomo-Pathology Department, CHC Montlegia, 4020 Liege, Belgium; veronique.jossa@chc.be

**Keywords:** sentinel lymph node, breast cancer, indocyanine green

## Abstract

**Simple Summary:**

This randomized study was conducted to evaluate sentinel lymph node biopsy with indocyanine green (ICG) compared with blue dye as a tracer in women with early breast cancer without any sign of lymph node invasion. ICG is a fluorescent tracer well known in medical practice for 50 years that is used as tracer of sentinel lymph nodes in numerous types of cancers other than breast cancer. This tracer is cheaper than radioactive tracers, with an easy learning curve.

**Abstract:**

Background: Indocyanine green (ICG) is a promising tracer for sentinel lymph node biopsy in early breast cancer. This randomized study was conducted to evaluate sentinel lymph node biopsy with ICG compared with blue dye as a tracer in woman with early breast cancer without any sign of lymph node invasion. Methods: Between January 2019 and November 2020, 240 consecutive women with early breast cancer were enrolled and randomized to sentinel lymph node biopsy using ICG or blue dye. The primary endpoint was the sentinel lymph node detection rate in both arms. Results: ICG was used in 121 patients and detected sentinel lymph nodes in all patients (detection rate, 100%; 95% CI: 96.9–100.0) while blue dye was used in 119 patients and detected sentinel lymph nodes in 116 patients (detection rate: 97.5%, 95% CI: 92.9–99.1). This analysis indicated the non-inferiority of ICG vs. blue dye tracer (90%CI: −1.9–6.9; *p* = 0.0009). Conclusion: ICG represents a new promising tracer to detect sentinel lymph nodes in early breast cancer with a detection rate similar to other conventional tracers, and is associated with easy learning and low cost. Our result suggest that this technique is a good alternative to avoid radioactive isotope manipulation.

## 1. Introduction

Since the 20th century, a less invasive method, the sentinel lymph node biopsy (SLNB) technique has been developed to decrease shoulder and arm morbidity [1] and has replaced the axillary lymph node dissection as the standard surgical approach to early breast cancers without pre-operative evidence of nodal infiltration [2,3].

The SLNB technique most frequently used worldwide is based on the radioisotope 99 m technetium either alone or in combination with blue dye. In a reference study published by Krag et al. in 2007, this combined method resulted in a high detection rate (97.2%) associated with a low false negative rate (9.8%) [4]. Since this study, guidelines have recommended a detection rate higher than 97% to be efficient for the SLN mapping method [5].

However, the use of a radioisotope tracer presents some disadvantages. First, the applicability of a radioisotope is limited to centers with nuclear medicine facilities and needs greater logistics and expensive equipment, including the tracer itself and the radioactive probe in the surgery room. Second, the injection of a radioisotope 4–6 h before surgery is a logistic challenge. Finally, the accessibility to radioactive technetium is not always easy, with a decrease in production sometimes leading to shortages of the tracer. Moreover, despite the limited radioactive doses received by surgeons and nurses in the surgery room, the use of a radioactive tracer is increasingly unpopular and tends to be replaced by a “greener” tracer.

A large number of developing countries and also some other countries in Asia do not use this mapping method and favor the use of patent blue dye alone, which is cost-effective [6,7]. In a previous study, we validated blue dye as a performing tracer in the hands of an experimented surgeon with a SLN detection rate of 97.4% in a community hospital center in Liege (CHC Liege, Belgium) [8].

Indocyanine green (ICG) is a fluorescent tracer that has been well known in medical practice over 50 years for intravenous injections to measure cardiac output or liver function [9]. ICG is also already being used as tracer of sentinel lymph nodes in numerous other types of cancers such as colon, stomach, prostate, cervix, endometrium and oropharynx cancers, needing only a near-infrared-camera in the surgical ward [10]. Allergic reactions were observed in less than 1 in 10,000 cases, which is much less than the rate for patent blue dye [11]. In the last decade, different studies, including two recent meta-analyses, have determined the detection rate of SLN by indocyanine green in combination with blue dye and/or 99 m technetium [12,13]. The large majority of these studies validated ICG as a performing tracer for sentinel lymph node detection with results as good as or even better than radioisotopes. However, the majority of these studies were performed in Asia and only few in Europe on relatively small cohorts of patients. Different centers in Asia and, particularly in China, have decided to use it in SLN mapping, while, in Europe, this technique has not yet replaced dual methods, and more and larger local studies are needed to prove the non-inferiority of the ICG mapping compared with classic methods in terms of the detection rate of SLN with a much lower cost and easier use.

Here, we performed a non-inferiority randomized trial to compare the ICG tracer with the patent blue dye. To the best of our knowledge, this is the first study using the ICG tracer not in combination with another tracer in patients with early breast cancer without any clinical signs of lymphatic node invasion.

## 2. Materials and Methods

### 2.1. Patients and Design

With a sample size of 120 in each tracer group and a sentinel lymph node detection rate of 97% in the blue dye tracer control group, the study achieved 73% power if the non-inferiority margin was set at −5%. Power calculations were performed using PASS 15 software.

Randomization was carried out in blocks of variable size defined randomly with sequence permutation as well as stratification according to the type of surgery (radical or conservative) performed using Excel software.

This prospective study enrolled 240 consecutive women with early breast cancer without any metastatic lymph nodes. Each patient underwent clinical and ultrasound examination of the axilla, and fine-needle aspiration of any suspicious nodes. The exclusion criteria were suspicious palpable axillary lymph nodes, suspicious senography nodes with positive cytology, a history of breast cancer, previous breast or axillary surgery, neoadjuvant chemotherapy and pregnancy. No loss to follow-up or exclusions were observed after randomization. Patients underwent sentinel lymph node biopsy followed by radical mastectomy or conservative surgery between January 2019 and November 2020 in a non-university hospital in Liege (CHC Liege, Belgium). Women did not receive neo-adjuvant chemotherapy and were randomized to the indocyanine green (ICG, *n* = 121) or patent blue dye cohorts (*n* = 119). Total axillary lymph node dissection was performed on the basis of positive frozen histological sections during surgery. Surgery was performed by the same surgeon in all patients. The surgeon has extensive experience in sentinel lymph node dissection with patent blue dye of more than 20 years and underwent a short training period in the use of indocyanine green before starting this study. The present study was conducted in accordance with the principles of the Declaration of Helsinki and was approved by the local ethics committee. All patients provided written informed consent.

### 2.2. Sentinel Lymph Node Mapping: Blue Dye Cohort

After general anesthesia, 2 mL of a solution of 2.5% diluted patent blue dye was injected under the dermis at the site of malignancy and also in the parenchyma in the 4 quadrants of the peritumoral area. After 1 min of manual massage, an incision was made directly above the axillary pyramid regardless of the breast surgery type (conservative or radical); this was a 4-cm-long antero-posterior incision located 5 cm from the arm root; blue stained nodes were removed. Generally, principal sentinel lymph (PSN) nodes were observed at the Berg 1 level, and less intense blue nodes called accessory sentinel nodes (ASN) could also be observed and biopsied at Berg level 1 and at the Berg level 2.

### 2.3. Indocyanine Green (ICG) Cohort

In the second cohort of patients, two intradermal injections of 0.1 mL of ICG each (0.5 mg of ICG injected per patient) at the peri-areolar level in front of the tumoral site were delivered. As with the blue dye tracer, massage was performed for 1 min and an incision was made directly above the axillary pyramid regardless of the breast surgery type (conservative or radical); this was a 4-cm-long antero-posterior incision located 5 cm from the arm root. From this moment, we were in complete darkness and only illuminated by the light emitted by the camera. To identify the lymphatic vessels that led to the sentinel nodes, we used the SPY fluorescence mode. Once the lymph nodes had been visualized, we switched to color segmented fluorescence mode to perform an excision of the sentinel lymph nodes only while respecting the surrounding tissue. As with the blue dye, principal and accessory lymph nodes were detected and biopsied at Berg level 1 and Berg level 2.

### 2.4. Non-Sentinel Lymph Nodes Detected

According to international guidelines, axillary clearance was performed in the case of positive sentinel lymph nodes. All of these nodes were labeled as non-SLN.

### 2.5. Anatomopathology

Each node was cut into sections of 2 mm and fixed in 4% formol. Sections of 5 µm were stained with hematoxylin–eosin and analyzed by an anatomopathologist to detect metastatic nodes. If all nodes per patient were negative, immunohistochemistry was performed with a specific antibody against cytokeratin CK-7 (Ventana USA, Westmoreland, PA, USA) to detect micro-metastasis.

### 2.6. Outcomes and Statistical Analyses

The primary endpoint of this prospective study was to prove the non-inferiority of the indocyanine green tracing method compared with the reference site method in terms of the sentinel lymph node detection rate.

The secondary endpoints were an evaluation of the number of SLNs biopsied for each patient, as well as the number of principal and accessory lymph nodes detected. The number of metastatic lymph nodes was also reported. Moreover, the duration of lymph node biopsy was also evaluated for both methods. Finally, the development of allergic reactions was collected in each cohort of patients.

Data were summarized as the mean and standard deviation (±SD) for quantitative variables, while frequency tables (number and percentages) were used for the categorical findings. Wilson’s 95% confidence intervals (95% CI) of the sentinel lymph node detection rate were estimated in both groups, and the non-inferiority analysis for the risk difference was conducted using the Farrington–Manning method. The results were considered significant at the 5% critical level. Data analysis was carried out using SAS (version 9.4).

## 3. Results

### 3.1. Patients’ and Tumors’ Characteristics

Sentinel lymph nodes were detected using the blue dye method alone for 119 women with early breast cancer and using the indocyanine green method for 121 other patients. The main characteristics of the patients and their tumors are summarized in Table 1. The mean age was 61 ± 11 years in each cohort, with more than 80% of patients being older than 50 years. Both cohorts of patients presented similar tumoral characteristics, with a majority of T1 and T2 breast cancers (90% in the blue dye cohort and 89% in the ICG cohort) and a Bloom score of 1 or 2 (86% in both groups). More than 70% of breast cancers presented an invasive ductal histology (77% in blue dye and 73% in ICG), while 6% of both groups presented an in situ component (seven patients in the blue group and eight patients in the ICG group). Moreover, conservative surgery was performed in 80% and 77% of patients in the blue dye and ICG cohorts, respectively.

### 3.2. Detection Rate and Characteristics of Sentinel Lymph Nodes

At least one sentinel lymph node was detected in 116 of 119 patients with the blue dye tracer alone, giving a detection rate of 97.5% (95% CI: 92.9–99.1), while indocyanine green allowed the detection of sentinel lymph nodes in all 121 patients (a detection rate of 100%, 95% CI: 96.9–100.0) (Table 2). The absolute difference between the ICG and blue dye tracer rates was 2.5% (90% CI: −1.9 to 6.9) (*p* = 0.0009 with margin = −5%). No clinical characteristics of three patients or of their tumors could explain the non-performance of blue dye in these patients.

This analysis indicated the non-inferiority of ICG vs. the blue dye tracer.

In total, 798 sentinel lymph nodes were removed, 360 from the 119 patients in the blue dye cohort and 438 from the 121 patients in the ICG cohort. A mean of 3.0 ± 1.5 and 3.6 ± 1.4 sentinel lymph nodes were removed per patient using, respectively, blue dye or ICG as the tracer. Moreover, SLNs were classified as principal or accessory sentinel lymph nodes according to their localization and the intensity of the detected tracer. In the blue dye cohort, 165 of the nodes removed corresponded to principal lymph nodes (PSN), while 195 were classified as accessory lymph nodes (ASN), as they were less labeled and were localized in the deeper part of the Berg 1 level or at the Berg 2 level. In the ICG cohort, of the 438 SLNs removed, 177 were classified as PSNs, while 261 were ASNs. The mean number of PSNs was 1.4 ± 0.7 with blue dye and 1.5 ± 0.8 with ICG. The mean number of ASNs was 1.9 ± 1.0 for blue dye and 2.3 ± 1.2 for ICG.

Pathological analyses of the SLNs showed that 48 patients presented at least one metastatic node among the 240 patients (20%, Table 3). Among them, 26 patients presented positive SLNs in the blue dye cohort and 22 in the ICG cohort. The majority of positive nodes were identified in the principal sentinel lymph nodes (42/48): 24 out of 26 patients were positive in the blue dye cohort and 18 out of 22 patients were in the ICG cohort. This means that the accessory lymph nodes alone allowed the detection of metastatic cancer in six patients (two in the blue dye cohort and four in the ICG cohort), corresponding to 12.5% of all metastatic patients detected (6/48).

The mean operation time from the beginning of ICG or blue dye injection to SLN biopsy completion was 20 min ± 12 for both methods in the case of radical mastectomy (Table 4). For conservative surgery, the operation time was shorter with blue dye (17 min ± 8) compared with ICG (20 min ± 12). However, this difference was not significant. No allergic reactions were observed in this study for blue dye or for ICG according to the Ring Messmer classification assessment (Table 4).

## 4. Discussion

For more than 20 years, sentinel lymph node biopsy has replaced axillary lymph node dissection as the standard of care to identify lymph node invasion in early breast cancer. International guidelines recommend a detection rate of SLN higher than 97% and the use of the radioisotope tracer 99 m technetium associated or not associated with blue dye [5]. However, the need for specific infrastructure and the associated cost of the radioisotope has led to the development of new tracers. The simplicity of use and the low cost of the fluorescence tracer indocyanine green led to its adoption for visualizing the lymphatic system and lymph nodes for breast cancer. In this last decade, different studies have proved that ICG mapping allowed a sentinel lymph node detection rate as high as the previously reported for a combination of blue dye and a radioisotope [13]. However, despite these promising results, the fluorescent mapping method has not been adopted in clinical practice, especially in European countries. The large majority of studies have come from Asia, where the most widespread tracer is blue dye alone, with a detection rate of around 87% according to a recent meta-analysis [12]. The ICG tracing method developed in Asia was easier to understand, as it improved the detection rate to 95–100%.

The superiority of another technique would be very difficult to prove.

Here, we performed a randomized non-inferiority trial including 121 early breast cancer patients for whom sentinel lymph nodes were mapped with ICG alone and 119 patients for whom the SLNs were mapped with blue dye alone. Regarding the blue dye injection sites, we used the same technique we have used for more than 20 years, which was validated as true in an already published cohort study [8]. Concerning ICG, we only reproduced the technique used in existing publications.

Blue dye mapping failed only in three patients among the 119, indicating a detection rate of 97.5%. This result is in line with our previous reference study performed with blue dye on a large cohort of patients, which reported a detection rate of 97.4% [8]. Moreover, this rate also corresponded to that obtained with combination methods with radioisotope tracers (97.2%) in the reference study NSABP B-32 published in 2007 [4]. This very high result with blue dye could question the necessity of finding a method other than ICG. However, the objective of this study was not to prove the superiority of ICG as compared with the reference method but to show its non-inferiority. With ICG mapping in 121 patients, sentinel lymph nodes were obtained from all of these patients, giving a detection rate of 100%.

Our results are in perfect agreement with the two significant meta-analyses recently published by Kedrzycki et al. [14] and by Thongvi-tokomarn and Polchai [15]. Indeed, as reported by these two studies, we have also demonstrated the non-inferiority of ICG compared with conventional techniques.

In opposition to blue dye, which gave discordant results in terms of the detection rate in the literature, which ranged from 70% to more than 95%, as in our center, almost all studies of ICG have shown a detection rate between 95% and 100% [3,12,13]. Indeed, in the two largest published cohort studies on 301 and 847 patients, the detection rates reported were 99% and 97%, respectively, with ICG [16,17]. With ICG, as with blue dye, sentinel lymph nodes could be observed through the axillary fat, avoiding blind exploration, as with a radioactive tracer. A major advantage of ICG as compared with blue dye is the visualization of fluorescent nodes on a screen in the surgery room. The direct observation on a screen is an important pedagogic tool for students and fellows. Direct visualization of SLNs, as with ICG, should allow more precise and anatomical surgery, preserving the architecture of the vascular and nervous system leading to reduced site injury.

No consensus was found regarding the number of sentinel lymph nodes to excise per patient; however, major studies showed the achievement of an average of three SLNs in common practice. In 2018, Qiu et al. showed that the mean number of SLNs excised with blue dye ranged from 1.0 to 2.3, while it ranged from 1.5 to 5.4 for ICG [3]. Compared with the radioisotope tracer, the mean number of SLNs removed with ICG was equal or higher to that with a radioisotope.

We were able to distinguish principal SLNs (PSNs) and accessory SLNs (ASNs). The biopsy of principal sentinel lymph nodes gave a mean number of 1.4 in the blue dye cohort and 1.5 in the ICG cohort. However, the addition of ASNs increased the mean number of SLNs removed to 3.0 with blue dye and 3.6 with ICG. The detection and biopsy of these ASNs were relevant, as they allowed the identification of infiltrated nodes in six patients, corresponding to 12.5% of all metastatic patients, while the PSNs were diagnosed as negative. Previous results showed an association between the excision of three sentinel nodes and an improvement in survival outcomes in patients with breast cancer [18]. In 2017, Kim et al. also showed similar results with a lower recurrence-free survival with one lymph node biopsied compared with patients with two or more lymph nodes removed [19].

The time needed for SLN surgery was similar with both mapping methods, namely around 18 min, demonstrating the ability to detect lymph nodes with a near-infrared camera. There is still a discordance in the concentrations, doses and locations of ICG injections among studies. Here, we injected 0.2 mL of ICG (0.5 mg of ICG per patient). Studies have shown that higher injected ICG concentrations lead to worse detectability due to fluorophore quenching or to unnecessary excision of nodes [11,20]. Standardization should be made in the guidelines.

Many advantages of ICG have been highlighted compared with other tracers. ICG is cheap, easier to use and does not require any specific equipment, except the near-infrared camera. The cost of this camera is higher than the neo-probe for the technetium method; however, as it is also used in many other applications in digestive, liver, plastic or gynecologic surgery, many surgical rooms are already equipped with it. These characteristics make it particularly attractive for regional centers that are unable to work with radioactive isotopes. In the recent meta-analysis by Goonawardena et al., the average cost of the radioisotope was estimated to be between USD331 and USD420 per patient, while that of ICG ranged between USD5 and USD111 per patient [13]. Of course, ICG is not cheaper compared with blue dye but, as explained above, ICG gives a higher detection rate and is easier to learn. Moreover, blue dye has been shown to induce allergic reactions of any grade in around 1% of patients [21] and severe reactions in 0.2% of the patients [4]. We can also consider the cosmetic results, where the patent blue dye shows “tattoos” more than 2 years after surgery, unlike ICG [22]. With ICG, all and severe reactions were observed in 0.34% and 0.05% of the patients, respectively [3]. In our study, no reaction were observed in the 240 patients enrolled in the study. At the end, our findings also agree with the assertions of the two recent meta-analysis [14,15] which stated that the use of ICG would not only be relevant in terms of SLNB efficacy, but would also eliminate the risks associated with ionizing radiation, skin tattooing and hypersensitivity reactions.

Because, in our study, we limited our surgery to SNLB if the frozen section was negative, we could not evaluate the false negative rate. However, Yin [23] gave us some interesting data concerning the false negative rate (FNR) in a meta-analysis. He compared the FNR of ICG versus RI and BD. Regarding ICG vs. RI, in a total of five studies with 1326 patients, the difference in FNR between ICG and RI was statistically significant, with a 50% reduced rate with ICG (OR = 0.46; 95% CI 0.23–0.91; *p* = 0.03). Regarding ICG vs. BD, in a total of eight studies including 683 patients, the overall FNR using ICG was 3.25%. This was lower than the FNR of using BD, which was 16.88%. In a logistic regression model, the results revealed a significant difference between the techniques (OR = 0.20; 95% CI: 0.08–0.48; *p* = 0.0004), which demonstrated that ICG had a lower FNR compared with BD. Moreover, in our previous cohort study [8] with BD alone, our FNR was, respectively, 9% and 4.9% if included only the principal sentinel nodes or all the sentinel nodes.

A major limitation of this study is that this is a mono-center study, and all surgical interventions were carried out by the same highly experienced surgeon. This could explain the higher detection rate observed with blue dye as compared with other studies. However, this study showed the high potential of indocyanine green to detect sentinel lymph nodes in early breast cancer. A large European multi-center confirmatory trial should be considered in order to decrease the use of radioactive isotopes in the medical field.

Researchers should discuss these results and how they can be interpreted from the perspective of previous studies and of the working hypotheses. The findings and their implications should be discussed in the broadest context possible. Future research directions may also be highlighted.

## 5. Conclusions

ICG represents a new promising tracer to detect sentinel lymph nodes in early breast cancer with a detection rate similar to other conventional tracers. Our results suggest that this technique is a good alternative in order to avoid radioactive isotope manipulations.

## Figures and Tables

**Table 1 cancers-14-00888-t001:** Characteristics of patients and tumors.

	Blue Dye Cohort *n* = 119 (%)	ICG Cohort *n* = 121 (%)
Total 240		
Age		
Years, mean ± SD:	61 ± 11	61 ± 11
<50 years	20 (17%)	22 (18%)
≥50 years	99 (83%)	99 (82%)
Laterality		
Left	65 (55%)	61(50%)
Right	54 (45%)	60 (50%)
Tumor size		
i.s.	7 (6%)	7 (6%)
T1	73 (61%)	84 (69%)
T2	35 (29%)	24 (20%)
T3	2 (2%)	6 (5%)
T4	2 (2%)	0 (0%)
Tumor localization		
Superior	81 (68%)	78 (64%)
Inferior	17 (14%)	20 (17%)
Median	12 (10%)	14 (12%)
Areolar or retro-areolar	8 (7%)	9 (7%)
NA	1 (1%)	0 (0%)
Bloom score		
1	48 (40%)	50 (41%)
2	55 (46%)	55 (45%)
3	8 (7%)	9 (7%)
Is	7 (6%)	7 (6%)
NA	1 (1%)	0 (0%)
Histological type		
In situ	7 (6%)	7 (6%)
Microinvasive	0 (0%)	1 (1%)
Invasive ductal	92 (77%)	88 (73%)
Invasive lobular	18 (15%)	21 (17%)
Other	2 (2%)	4 (3%)
Surgery type		
Radical	24 (20%)	28 (23%)
Conservative	95 (80%)	93 (77%)

ICG, Indocyanine green; SD, standard deviation; NST, non-specific type.

**Table 2 cancers-14-00888-t002:** Lymph node detection.

	Blue Dye *n* = 119	ICG *n* = 121
SLN detection rate	116 (97.5%)	121 (100%)
Total number of SLNs	360	438
Principal lymph nodes	165	177
Accessory lymph nodes	195	261
Mean number of SLN (±SD)	3.0 ± 1.5	3.6 ± 1.4
Principal lymph nodes	1.4 ± 0.7	1.5 ± 0.8
Accessory lymph nodes	1.9 ± 1.0	2.3 ± 1.2

SLN, sentinel lymph node; SD, standard deviation; ICG, indocyanine green.

**Table 3 cancers-14-00888-t003:** Positive lymph nodes.

	Blue Dye *n* = 119	ICG *n* = 121	Total *n* = 240
SLN-positive patients (detection rate)	26 (21.8%)	22 (18.2%)	48 (20%)
PSN-positive	24 (20.2%)	18 (14.9%)	42 (17.5%)
ASN-positive	13 (11.2%)	11 (9.1%)	24 (10%)
ASN-positive/SNP negative	2 (1.7%)	4 (3.3%)	6 (2.5%)

SLN, sentinel lymph node; PSN, principal sentinel lymph node; ASN, accessory sentinel lymph node; ICG, indocyanine green.

**Table 4 cancers-14-00888-t004:** Duration of SLNB and allergic reactions.

	Blue Dye *n* = 119	ICG *n* = 121
Time of SLN surgery (min, mean ± SD)		
Radical mastectomy	20 ± 12	20 ± 12
Conservative surgery	17 ± 8	20 ± 12
Allergic reactions	0/119 (0%)	0/121 (0%)

SLN, sentinel lymph node; SD, standard deviation; ICG, indocyanine green.

## Data Availability

The datasets used or analyzed during the current study are available from the corresponding author on reasonable request.

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
