# Peer review of "A Randomized Prospective Non-Inferiority Trial of Sentinel Lymph Node Biopsy in Early Breast Cancer: Blue Dye Compared with Indocyanine Green Fluorescence Tracer"

_cancers, 2022, doi:10.3390/cancers14040888_

Round 1

Reviewer 1 Report

A more detailed description of the ICG technique is needed. Please specify which infra-red camera is used.

Does the used camera allow visible spectrum and ICG imaging fusion in real-time?

The injection site is different in both arms. This is a major issue about the methodology that may explain some differences in sentinel lymph node detection. The authors should explain in the discussion why they use those different approaches.

The discussion can be improved by an exhaustive revision of published articles. This revision must include the recently published meta-analysis (Kedrzycki et al., 2021; Thongvitokomarn & Polchai, 2020)s and relevant articles(Jung et al., 2014).

Author Response

  • A more detailed description of the ICG technique is needed. Please specify which infra-red camera is used.

We use a Spy Portable Handheld Imager (SPY-PHI) from Stryker-Novadaq

After injecting the ICG and performed a one minute breast massage, we incise the skin directly above the axillary pyramid regardless of the breast surgery (conservative or radical): this is a 4 cm long antero-posterior incision, located 5 cm from the arm root. From this moment, we are in complete darkness only illuminated by the light emitted by the camera. To identify the lymphatic vessels that lead to the sentinel nodes, we use the SPY fluorescence mode. Once the lymph nodes are visualized, we switch to Color Segmented Fluorescence Mode to only perform an excision of the sentinel lymph nodes while respecting the surrounding tissue

  • Does the used camera allow visible spectrum and ICG imaging fusion in real-time?

Yes, the used camera allows visible spectrum and ICG imaging fusion in real-time

  • The injection site is different in both arms. This is a major issue about the methodology that may explain some differences in sentinel lymph node detection. The authors should explain in the discussion why they use those different approaches.

Regarding Blue Dye injection sites, we have been successful in using the patent blue (2ml) injection method under the dermis at the site of malignancy and also in the parenchyma at the 4 quadrants of the peritumoral area for more than 20 years. We also evaluated the performance of this technique on a series of 456 patients and recently published it (Olivier et al. Gland Surg. 2021).

Under these conditions, we obtained a detection rate of 97.4%, which validates the technique and the tracer.

Concerning ICG, we only reproduced the technique used in existing publications (at the periaerolar level in front of the tumor).

However, before starting our randomized study, we evaluated this procedure in a set of patients in our institution, by combining the two tracers and their respective injection sites. In addition, this preliminary experiences served as a learning curve.

  • The discussion can be improved by an exhaustive revision of published articles. This revision must include the recently published meta-analysis (Kedrzycki et al., 2021; Thongvitokomarn & Polchai, 2020) and relevant articles (Jung et al., 2014).

Our results are in perfect agreement with the two significant meta-analyses recently published respectively by Kedrzycki et al., 2021 and by Thongvitokomarn & Polchai, 2020. Indeed, as similarly reported by these 2 studies, we have also demonstrated the non-inferiority of ICG compared to conventional techniques.

This allows us to validate the fluorescence technique in the SLNB for early breast cancer.

Our feeling also agrees with the assertions of these two authors when they state that the use of ICG would not only be relevant in terms of SLNB efficacy, but would eliminate the risks associated with ionizing radiation, skin tattooing, and hypersensitivity reactions. (Kedrzycki et al., 2021)

To allow an everywhere accessibility to a reliable technique, it seems essential to go straight to the point and to avoid too sophisticated approach as dual or multimodal method (MMM) described by (Jung et al., 2014).

Reviewer 2 Report

The study is a non-inferiority study comparing 2 types of sentinel biopsies.

The study is well designed. The authors have to describe the types of incisions to the axilla (location, length), and if the mastectomy incision was used for sentinel in respective cases. 

Then we need a description of the ICG-camera (type etc) and how this was used. It is just stated "monitor".

Then they have to present the preoperative axillary evaluation (technique and assessment). In the known SLN techniques the metastatic nodes are missed if the metastasis is to big. Is there data on this topic available for ICG? How is the rate of false negative SLN in the applied techniques, is there available data? Is a follow up planned to assess recurrence status?  

Please correct the first sentence of introduction. 

Author Response

The study is a non-inferiority study comparing 2 types of sentinel biopsies.

The study is well designed. The authors have to describe the types of incisions to the axilla (location, length), and if the mastectomy incision was used for sentinel in respective cases. 

After injecting the ICG and performed a one minute breast massage, we incise the skin directly above the axillary pyramid regardless of the breast surgery (conservative or radical): this is a 4 cm long antero-posterior incision, located 5 cm from the arm root. From this moment, we are in complete darkness only illuminated by the light emitted by the camera. To identify the lymphatic vessels that lead to the sentinel nodes, we use the SPY fluorescence mode. Once the lymph nodes are visualized, we switch to Color Segmented Fluorescence Mode to only perform an excision of the sentinel lymph nodes while respecting the surrounding tissue

  • Then we need a description of the ICG-camera (type etc) and how this was used. It is just stated "monitor".

Supra.

We use a full system Spy Portable Handheld Imager (SPY-PHI) from Stryker-Novadaq.

The Spy camera allow visible spectrum and ICG imaging fusion in real-time.

  • Then they have to present the preoperative axillary evaluation (technique and assessment).

Each patient got clinical and ultra sound examination of the axilla, and fine-needle aspiration of any suspicious nodes.

The exclusion criteria were suspicious palpable axillary lymph nodes; suspicious senography nodes with positive cytology; history of breast cancer; previous breast or axillary surgery; neoadjuvant chemotherapy; and pregnancy.

  • In the known SLN techniques the metastatic nodes are missed if the metastasis is too big. Is there data on this topic available for ICG?

At our knowledge, no data is available concerning this question. Anyway, this is an exclusion criteria in our study.

  • How is the rate of false negative SLN in the applied techniques, is there available data?

Because in our study, we limited our surgery to SNLB if the frozen section is negative, we cannot evaluate the false negative rate.

But Yin (Rui Yin Oncology letter December 15, 2020) gives us some interesting data concerning false negative rate (FNR) in a meta-analysis study. He compares the FNR of ICG versus RI and BD.

Regarding ICG Vs RI, from a total of five studies with 1,326 patients,  the difference in FNR between ICG and RI was statistically significant with a 50% reduced rate with ICG (OR=0.46; 95% CI 0.23-0.91; P=0.03).

Regarding ICG vs BD, from a total of eight studies including 683 patients, the overall FNR using ICG was 3.25%. This was lower than the FNR of using BD, which was 16.88%. Using a logistic regression model, the results revealed a significant difference between the techniques (OR=0.20; 95% CI, 0.08-0.48; P=0.0004), which demonstrated that ICG had a lower FNR compared with BD.

Moreover in our previous cohort study (Olivier et al. Gland Surg. 2021), with BD alone, our FNR was respectively 9% and 4.9% if we took into account only the principal sentinel node or all the sentinel nodes.

  • Is a follow up planned to assess recurrence status?  

Like in usual practice, senology, imagery, gynecology, oncology.

Reviewer 3 Report

The manuscript by Coibion et al. indicates that Indocyanine Green (ICG) dye is a comparable approach to blue dye for sentinel lymph node biopsy of early stage of breast cancer patients. Authors claim that ICG shows higher detection rates of sentinel lymph node relative to blue dye using a single cohort patient-randomized trials, although statistical significances between detection rates of ICG (100%) and blue dye (97.5%) was not shown, which seems not significant. Interestingly, blue dye showed a higher sensitivity for positive lymph node detection (21.8%) than those of IGC (18.2%), which should also be considered in results and discussion to withdraw conclusion. Clinical significance of IGC may include low cost and easy learning of IGC compared to blue dye, which was not directly analyzed in this study. Major concern is that the detailed information about how to assess adverse effect of ICG during trials was not addressed, thus it would enhance quality of this patient study and help reviewers/readers to conclude whether ICG is a good alternative of blue dye for sentinel lymph node biopsy.  

Author Response

The manuscript by Coibion et al. indicates that Indocyanine Green (ICG) dye is a comparable approach to blue dye for sentinel lymph node biopsy of early stage of breast cancer patients.

  • Authors claim that ICG shows higher detection rates of sentinel lymph node relative to blue dye using a single cohort patient-randomized trials, although statistical significances between detection rates of ICG (100%) and blue dye (97.5%) was not shown, which seems not significant.

This is a non-inferiority study. Authors claim that ICG is as good as Blue dye and analyses indicate a non-inferiority of ICG vs blue dye tracer (90%CI: -1.9 - 6.9; p=0.0009).

  • Interestingly, blue dye showed a higher sensitivity for positive lymph node detection (21.8%) than those of IGC (18.2%), which should also be considered in results and discussion to withdraw conclusion.

The difference between the proportions is not significant according a 2- sample test for equality of proportions without continuity correction. The difference between the proportions is not significant:

 > prop.test(c(26, 22), c(119, 121), alternative = c("two.sided"), conf.level=0.95, correct = FALSE)

data:  c(26, 22) out of c(119, 121)

X-squared = 0.5042, df = 1, p-value = 0.4777

alternative hypothesis: two.sided

95 percent confidence interval:

-0.06449804  0.13783646

sample estimates:

   prop 1    prop 2

0.2184874 0.1818182

  • Clinical significance of IGC may include low cost and easy learning of IGC compared to blue dye, which was not directly analyzed in this study.

Of course ICG was not cheaper compared to blue dye because of the need of an expensive near-infrared camera. But, note this device is also used in many other applications in surgery rooms thus explaining that many hospitals are already equipped with it.

ICG technique, because of the use of big screen shared with all the surgical team, is convivial and much easier to learn than BD technique.

  • Major concern is that the detailed information about how to assess adverse effect of ICG during trials was not addressed, thus it would enhance quality of this patient study and help reviewers/readers to conclude whether ICG is a good alternative of blue dye for sentinel lymph node biopsy.  

We use the Ring Messmer classification to assess eventual allergic adverse effect of ICG /Blue dye during this study. Fortunately we did not encounter any side effects regarding dyes during this study. In the literature blue dye has been shown to induce allergic reactions in around 1% for any grade [19] and 0.2% for severe reactions of the patients [4].

Concerning ICG, allergic reactions were observed in less than 1 for 10000 cases which is much less than patent blue dye [11]
